# RGSB-UNet: Hybrid Deep Learning Framework for Tumour Segmentation in Digital Pathology Images

**DOI:** 10.3390/bioengineering10080957

**Published:** 2023-08-12

**Authors:** Tengfei Zhao, Chong Fu, Ming Tie, Chiu-Wing Sham, Hongfeng Ma

**Affiliations:** 1School of Computer Science and Engineering, Northeastern University, Shenyang 110819, China; 2Engineering Research Center of Security Technology of Complex Network System, Ministry of Education, Shenyang 110819, China; 3Key Laboratory of Intelligent Computing in Medical Image, Ministry of Education, Northeastern University, Shenyang 110819, China; 4Science and Technology on Space Physics Laboratory, Beijing 100076, China; 5School of Computer Science, The University of Auckland, Auckland 1142, New Zealand; 6Dopamine Group Ltd., Auckland 1542, New Zealand

**Keywords:** hybrid deep learning framework, tumour segmentation, whole slide image, Residual-Ghost-SN, bottleneck transformer

## Abstract

Colorectal cancer (CRC) is a prevalent gastrointestinal tumour with high incidence and mortality rates. Early screening for CRC can improve cure rates and reduce mortality. Recently, deep convolution neural network (CNN)-based pathological image diagnosis has been intensively studied to meet the challenge of time-consuming and labour-intense manual analysis of high-resolution whole slide images (WSIs). Despite the achievements made, deep CNN-based methods still suffer from some limitations, and the fundamental problem is that they cannot capture global features. To address this issue, we propose a hybrid deep learning framework (RGSB-UNet) for automatic tumour segmentation in WSIs. The framework adopts a UNet architecture that consists of the newly-designed residual ghost block with switchable normalization (RGS) and the bottleneck transformer (BoT) for downsampling to extract refined features, and the transposed convolution and 1 × 1 convolution with ReLU for upsampling to restore the feature map resolution to that of the original image. The proposed framework combines the advantages of the spatial-local correlation of CNNs and the long-distance feature dependencies of BoT, ensuring its capacity of extracting more refined features and robustness to varying batch sizes. Additionally, we consider a class-wise dice loss (CDL) function to train the segmentation network. The proposed network achieves state-of-the-art segmentation performance under small batch sizes. Experimental results on DigestPath2019 and GlaS datasets demonstrate that our proposed model produces superior evaluation scores and state-of-the-art segmentation results.

## 1. Introduction

Colorectal cancer (CRC) is a gastrointestinal tumour that has a higher incidence and mortality rate than common tumours [1,2]. However, early screening with colonoscopy followed by pathological biopsy can significantly reduce the mortality rate [3]. Pathology is considered the gold standard for distinguishing between benign and malignant CRCs. During a diagnosis, physicians analyse the tumour’s condition by observing the H&E-stained pathological section, drawing on their clinical expertise [4].

The use of high-resolution, large-scale whole slide images (WSIs) has become a routine diagnostic method with the rapid development of image scanning techniques [5]. WSI technology has great potential for developing and using algorithms for pathological diagnosis [6]. WSIs are widely used for digital pathology analysis, particularly in clinical practice [7]. However, the large size of WSIs can make manual analysis by pathologists time-consuming, and the unavoidable cognitive biases can lead to varying diagnoses.

CRC segmentation in whole slide images presents a unique set of implementation challenges due to the high-resolution and large size of these images, including gigapixel-scale data, computational resources, data handling and preprocessing, and integration with clinical workflow. Addressing these challenges often involves a combination of advanced image processing techniques, deep learning architectures tailored for large images, efficient data handling methods, and collaboration between medical experts and computer scientists. Overcoming these challenges is critical to harness the full potential of whole slide image segmentation in improving the accuracy and efficiency of colon cancer diagnosis and treatment planning.

In recent years, deep learning-based approaches [8] have been widely applied to histopathology image analysis, achieving remarkable results. In [9], Xu et al., proposed a deep learning method based on convolutional neural networks (CNNs) to automatically segment and classify epithelial and stromal regions in histopathology images. In [10], Liu et al., proposed a framework for the automatic detection and localization of breast tumours. In [11], Wang et al. proposed a deep CNN method to automatically identify the tumour in lung cancer images, using the shape feature to predict survival outcomes. In [12], Johnson et al. used Mask-RCNN to segment the nuclei in pathology images. In [13], Fan et al. proposed an improved deep learning method based on a classification pipeline to detect cancer metastases in WSI. In [14], Cho et al. proposed a deep neural network with scribbles for interactive pathology image segmentation. In [15], Zhai et al. proposed deep neural network guided by an attention mechanism for segmentation of liver pathology images. In [16], Deng et al. proposed a interpretable multi-modal image registration network based on disentangled convolutional sparse coding to solve the problem of lack of interpretability. In [17], Jin et al. proposed a two-stage deep learning system named iERM to provide accurate automatic grading of epiretinal membranes for clinical practice. In [18], Xiong et al. proposed DCGNN, a novel single-stage 3D object detection network based on density clustering and graph neural networks. DCGNN utlized density clustering ball query to partition the point cloud space and exploits local and global relationships by graph neural networks.

While histopathological image analysis has shown remarkable results, few studies have investigated deep learning-based methods for CRC tissue segmentation, particularly in WSIs. In [19], Qaiser et al. introduced two versions of our tumour segmentation method: one aimed at achieving faster processing while maintaining accuracy, and the other focused on achieving higher accuracy. The faster version relied on selecting representative image patches from a convolutional neural network (CNN) and classifying the patches by quantifying the difference between the exemplars’ persistent homology profiles (PHPs) and the input image patch. In contrast, the more accurate version combined the PHPs with high-level CNN features and utilized a multi-stage ensemble strategy to label image patches. In [20], Zhu et al. proposed an adversarial context-aware and appearance consistency UNet (CAC-UNet) for segmentation and classification tasks, and achieved first place in the DigestPath2019 challenge. In [21], Feng et al. employed a UNet with a VGG backbone for WSI-based colorectal tumour segmentation, and achieved second place in the DigestPath2019 challenge.

Despite the remarkable results achieved by the methods mentioned above, several challenges still persist, including fewer public CRC datasets with expert annotations and difficulty accurately segmenting the refined boundary of the tumour, impeding further research on CRC tissue segmentation. Additionally, most existing deep learning frameworks rely on convolutional stacking, which reduces local redundancy but fails to capture global dependencies owing to the limited receptive field [22]. By contrast, transformers can capture long-distance dependencies through self-attention. However, excessive visual-semantic alignment may lead to redundancy in token representation, making it necessary to balance global dependency and local specificity when designing deep learning models.

This study proposes a hybrid deep learning framework for segmenting the CRC tumour in WSIs with a focus on refining the boundary segmentation and addressing network stability under small batch sizes. The proposed encoder–decoder architecture utilizes a newly designed encoder that includes residual ghost blocks with switchable normalization (RGS) and a bottleneck transformer block (BoT) for downsampling, while the decoder employs transpose convolution for upsampling [23,24,25,26,27]. By leveraging the benefits of CNNs and the transformer, the proposed encoder uses RGS and BoT as downsampling operations to extract more refined features from input images. The operation extracts local information, and the multi-head self-attention (MHSA) in the BoT models global dependency [27]. Experimental results demonstrate that the proposed model can accurately segment the tumour and produce a more refined boundary, leading to improved segmentation accuracy under small batch sizes. The primary contributions of our study are outlined below:We propose a deep hybrid network that combines a transformer and CNN for automatic tumour region segmentation in pathology images of the colon.A newly-designed feature extraction block RGS is presented. The block can adaptively determine the optimal combination of normalizers for each layer, making our model robust to varying batch sizes.Our novel hybrid backbone encoder, which includes RGS and BoT blocks, can extract more refined features.Experimental results demonstrate that the proposed RGSB-UNet achieves higher evaluation scores and produces finer segmentation results than state-of-the-art segmentation methods under small batch sizes.

The remainder of this paper is structured as follows. In Section 2, we present the proposed network architecture. Section 3 describes the datasets and evaluation criteria used in our experiments, while Section 4 presents our experimental results. Finally, in Section 5, we summarize the study results and suggest potential avenues for future research.

## 2. Proposed Method

### 2.1. Network Architecture

Our proposed deep learning framework for colon pathology WSI analysis is illustrated in Figure 1. As shown in Figure 2, to extract relevant features from original images, we start with 512×512×3 image patches using dense cropping methods. The encoder includes a novel downsampling operation that combines RGS and BoT blocks as the feature extraction backbone. The details of the design of the encoder and decoder, GBS, RGS, and BoT will be discussed below.

#### 2.1.1. Encoder and Decoder

In order to extract an efficient set of features, we use two 3 × 3 convolutions with batch normalization and ReLU, following a max pooling for downsampling, and devise a new residual ghost network, embedding a BoT at the end of the encoder as part of the encoder in our network architecture. The network employs four downsampling modules, each utilizing a different number of residual ghost blocks. As shown in Figure 2, the first downsampling module uses a 3 × 3 max pooling (MP) and a residual ghost block; the second and third downsampling modules use two and three stacked residual ghost blocks, respectively. By leveraging the ghost convolution technique, our network can generate rich feature maps using significantly fewer input features than traditional convolution methods, which improves the computational efficiency of our encoder. Additionally, the stability of our network is enhanced by the ability to select optimal combinations of different normalizers for each normalization layer, resulting in an accuracy that is not impacted by batch size. The fourth downsampling module incorporates a BoT block and a 2×2 average pooling (AP), which significantly boosts the extraction of refined features. Each downsampling module reduces the input spatial resolution by a factor of two.

The decoder is composed of four upsampling modules that utilize a transposed convolution and a 1 × 1 convolution with ReLU [28], increasing the input spatial resolution by a factor of two. The concatenate block concatenates the skip and output features of Tconv-ReLU; this operation attaches more local information extracted from different layers of the encoder directly into their corresponding decoder layers at the same level, which adds detailed information to the general area of the target of judgment. Further elaboration on the RGS and BoT components will be provided in subsequent subsections.

#### 2.1.2. Ghost Block with Switchable Normalization

Our proposed Ghost-Block-SN architecture is presented in Figure 3, which utilizes the Ghost-Block to generate more representative features at a lower computational cost. The Ghost-Block firstly employs traditional convolution to generate intrinsic feature maps and then utilizes cost-effective linear operations to expand the features and channels. The computational cost of linear operations on feature maps is much lower than traditional convolution, making the block more efficient than other existing efficient methods. The size of the primary convolution kernel in Ghost-Block is customizable, and we used a 1×1 point-wise convolution in our study. A BN layer is introduced after each Ghost-Block in Residual-Ghost-Block, which provides stability and speeds up the training process.

However, the performance of Ghost-Block-BN is restricted by the batch size as BN uses a single normalizer throughout the network, which can be unstable and degrade accuracy under small batch sizes. To overcome this issue, we incorporated switchable normalization (SN) [29], a technique that is robust to a wide range of batch sizes. SN measures channel-wise, layer-wise, and minibatch-wise statistics using BN [30], instance normalization (IN) [31], and layer normalization (LN) [32], respectively, and learns their important weights to find their optimal combination, ensuring network stability and accuracy in the case of small batch sizes.

#### 2.1.3. Residual Ghost Block with Switchable Normalization

As shown in Figure 4a, our RGS is constructed by incorporating the above presented GBS with a residual bottleneck, which is the fundamental building block of a ResNet [23], due to its exceptional performance. The core concept behind a residual block is to reformulate the layers as learning residual functions with respect to the layer inputs, rather than learning unreferenced functions. Compared to ResNet-50, our encoder employs fewer building units, boosting the computational efficiency. Moreover, the proposed RGS is highly robust and can handle a wide range of batch sizes.

#### 2.1.4. Bottleneck Transformer

Figure 4b shows the bottleneck transformer (BoT), an important block in the proposed hybrid network, which uses multi-head self-attention (MHSA) to replace the 3 × 3 convolution compared with RGS. The BoT is embedded in the last layer of the encoder. As is known, the self-attention (Figure 5a) can process and aggregate the information in the feature maps to complement the CNN handle long-distance dependencies. Particularly, the self-attention in MHSA can help the network better understand the relationships between different regions and improve the accuracy of segmentation when working with highly detailed images. In addition, as shown in Figure 5b, the MHSA with sufficient heads is at least as expressive as any convolutional layer [27]. The MHSA produces multiple attention maps and embedding features from an image to encode rich information, enhancing the deep model’s robustness towards representation learning. Benefiting from the MHSA, the BoT block can help the network to boost the segmentation performance.

### 2.2. Loss Function

Dice loss is leveraged as a standard loss function in image segmentation tasks and indicates the difference between the predicted and ground-truth mask [33]. However, there are still some limitations when employing this function. For instance, there is no segmenting target, and the dice loss is 0. Clearly, the dice loss function receives no punishment when predicting a false positive.

To address this issue, the improved class-wise dice loss function is leveraged to compute the background and lesion segmentation dice similarity coefficients (DSCs) for benign and malignant images, respectively [21]. The improved loss function can effectively reduce false positives, including its practicality for clinical applications. The improved class-wise dice loss (CDL) function is described by
(1)$$L_{\mathrm{CDL}}=1-\sum_{i}^{N}\left(y_{p}\frac{y_{i}\hat{y_{i}}}{y_{i}+\hat{y_{i}}}+\frac{\left(1-y_{p}\right)\left(1-y_{i}\right)\left(1-\hat{y_{i}}\right)+\epsilon }{\left(1-y_{i}\right)+\left(1-\hat{y_{i}}\right)+\epsilon }\right)\;,$$
where $y_{i}$ is the binary label of pixel *i*, $\hat{y_{i}}$ is the predicted probability, and *N* is the total number of pixels in a patch. ϵ is a small number to avoid the denominator becoming 0.

The presence of a lesion area determines the patch label ($y_{p}$). The CDL function can alleviate pixel-level class imbalance, resulting in an all-zero mask when training negative samples.

## 3. Evaluation and Datasets

### 3.1. Evaluation

We use the *DSC*, Jaccard Index (*JI*), and relative volume difference (*RVD*) to measure the segmentation performance of our proposed model [34]. The *DSC* measures the similarity between the network segmentation results when using the proposed method and the gold standard mask in image segmentation. *DSC*, *JI*, and *RVD* are defined as
(2)$$\mathrm{DSC}=\frac{2|Y_{A}\cap Y_{P}|}{\left|Y_{A}\right|+\left|Y_{P}\right|},$$
(3)$$\mathrm{JI}=\frac{\left|Y_{A}\cap Y_{P}\right|}{\left|Y_{A}\right|+\left|Y_{P}\right|-\left|Y_{A}\cap Y_{P}\right|},$$
and
(4)$$\mathrm{RVD}=\frac{\left|Y_{P}\right|-\left|Y_{A}\right|}{\left|Y_{A}\right|},$$
where $Y_{A}$ is the set of lesion pixels in the annotation, and $Y_{P}$ is the corresponding set of lesion pixels in the segmentation result.

We use pixel accuracy (*PA*) and area under the curve (*AUC*) to measure the classification performance of our proposed model. *AUC* is defined as the area of the receiver operating characteristic (ROC) curve, determined by the true positive rate (*TPR*) and false positive rate (*FPR*). *TPR*, *FPR*, and *Precision* are defined as follows:(5)$$\mathrm{TPR}=\frac{TP}{TP+FN},$$
(6)$$\mathrm{FPR}=\frac{FP}{FP+TN},$$
and
(7)$$\mathrm{Precision}=\frac{TP}{TP+FP},$$
where *TP*, *FP*, *TN*, and *FN* are true positives, false positives, true negatives, and false negatives, respectively.

*AUC* and *PA* are defined as
(8)$$\mathrm{AUC}=\int_{x=0}^{1}TPR\left(FPR^{-1}\left(x\right)\right)dx=P\left(X_{1}>X_{0}\right)$$
and
(9)$$\mathrm{PA}=\frac{TP+TN}{TP+TN+FP+FN},$$
where $X_{0}$ and $X_{1}$ are the scores for the negative and positive instances, respectively.

### 3.2. Datasets and Implementation

We trained the proposed network on the DigestPath2019 [35] gland segmentation (GlaS) [36] datasets. In these datasets, numerous expert-level annotations on digestive system pathological images are available, which will substantially advance research on automatic segmentation and classification of pathological tissues.

The DigestPath2019 dataset contains positive and negative samples of 872 tissue slices from 476 patients. The average size of a tissue slice is 3000 × 3000. The training set comprises 660 images from 324 patients, from which 250 images from 93 patients are annotated by pathologists. The positive training samples contain 250 tissue images from 93 WSIs, with pixel-level annotation, where 0 indicates the background and 255 indicates the foreground (malignant lesion). Some samples cropped from WSI are shown in Figure 6. The negative training samples contain 410 tissue images from 231 WSIs. These negative images have no annotation because they have no malignant lesions. The entry to DigestPath2019 competition has closed and the official test set is not publicly accessible. To address this issue, we remake a balanced test set by randomly selecting 108 samples with a 54:54 positive to negative ratio from the original training set. We retrained all the compared models on the DigestPath2019 dataset using their original code, and the test set images are not used in training. Defining an objective criteria for distinguishing between benign (negative) and malignant (positive) lesions is difficult. To make it easier for academic research, according to the WHO classification of digestive system tumours, we regarded the following lesions as malignant: high-grade intraepithelial neoplasia and adenocarcinoma, including papillary adenocarcinoma, mucinous adenocarcinoma, poorly cohesive carcinoma, and signet ring cell carcinoma. Low-grade intraepithelial neoplasia and severe inflammation are not included in the dataset because they are generally difficult for pathologists to detect.

The GlaS dataset consists of 165 tissue slices containing both positive and negative samples. The GlaS dataset contains a training set of 85 samples from which we selected 17 samples as the validation data. The dataset offers two different test sets, testA and testB, consisting of 60 and 20 samples, respectively. We used the validation set to select the optimal model and all the performance evaluations are carried out on the joining of testA and testB. Glands are vital histological structures found in various organ systems, serving as the primary mechanism for protein and carbohydrate secretion. Adenocarcinomas, which are malignant tumors originating from glandular epithelium, have been identified as the most prevalent form of cancer. Pathologists routinely rely on gland morphology to assess the malignancy level of several adenocarcinomas, such as those affecting the prostate, breast, lung, and colon. Accurately segmenting glands is often a crucial step in obtaining reliable morphological statistics. However, this task is inherently challenging due to the significant variation in glandular morphology across different histologic grades. Most studies to date have primarily focused on gland segmentation in healthy or benign samples, with limited attention given to intermediate or high-grade cancer. Additionally, these studies often optimize their methods for specific datasets.

The simulations were run on a station equipped with an NVIDIA GeForce RTX 3090 GPU and Intel(R) Xeon(R) CPU E5-2680v4×2. We augmented the training data during training. Table 1 lists the detailed hyperparameters of the proposed framework. We embarked on an iterative journey of manual tuning, wherein we systematically explored and fine-tuned various hyperparameters within our framework. By meticulously adjusting parameters such as learning rates, batch sizes, and model architecture, we meticulously tracked the impact of each modification on the overall performance metrics. This exhaustive process allowed us to discover the optimal combination of hyperparameters, leading to a highly refined and efficient version of our framework that exhibits superior accuracy and generalization on diverse datasets.

## 4. Experimental Results

Table 2 shows the results of the ablation study, which demonstrate the performance gains when integrating different blocks into UNet, including residual block (RSB), residual ghost block (RGB), RGS, and BoT. Especially, our proposed RGSB-UNet achieves the highest DSC score of 0.8336. We further analyze the performance of different batch sizes and MHSA head numbers based on RGSB-UNet. As is shown in Table 3, the proposed network maintains high performance even with small batch sizes. We tried different small batch sizes in our experiments. We prove that batch size is no longer a strict limitation for the proposed network. In addition, the head numbers of MHSA impact the performance of the proposed network. We have tried different numbers of heads for the MHSA in the proposed network to search for the best results, and our network achieved optimal performance when the heads are four. When integrating RGS and BoT together to the UNet, the segmentation model produces the best performance, which indicates that these blocks can improve the performance of pathology image segmentation.

Table 4 compares the performance of the proposed and other popular models in terms of six metrics on the DigestPath2019 dataset; the numbers in bold indicate the best results for each metric. As can be seen from this table, under a small batch of two, our proposed model achieves the highest DSC, PA, JI, and Precision; it also achieves the second best RVD and AUC. Furthermore, although DeepLab with Xception backbone outperforms other models in terms of RVD, and the CAC-UNet (first place) achieves the highest AUC, our model performs significantly better in the other three metrics. In Figure 7, we illustrate the results of tumour segmentation on the sample images and compare them with that of [20,21,37,38,39,40,41,42,43,44,45,46]. As shown in this figure, the mask predicted by the proposed network is extremely close to the ground truth. Compared with other leading networks, our proposed network can successfully segment tumour regions with nearly overlapping margins, indicated in the red boxes. Overall, our proposed model can capture more refined features and achieve state-of-the-art accuracy in tumour segmentation tasks.

To demonstrate our proposed method’s generalizability and its performance in different contexts, we use the GlaS dataset to verify the network. As shown in Table 5, our proposed model achieves the highest scores in state-of-the-art accuracy in gland segmentation tasks. Figure 8 shows the results of gland segmentation on the test set and compares them with [21,37,38,39,40,41,42,43,44,45,46]. As shown from this figure, compared with other leading works, our proposed network can significantly segment gland boundaries, as indicated in the red box. Our idea can be directly applied to a computer-aided pathological diagnosis system to reduce the workload of pathologists.

## 5. Conclusions

In this paper, we propose a hybrid deep learning framework for segmenting tumours in WSIs. Our model employs an encoder–decoder architecture, with a newly designed RGS block and a BoT block in the decoder part. These blocks are implemented to capture more refined features and improve network stability, particularly when working with small batch sizes. To evaluate the performance of our approach, we conducted extensive experiments on the DigestPath2019 and GlaS datasets, and the results indicate that our model achieved state-of-the-art segmentation accuracy.

Our proposed framework is generic and can be easily applied to other histopathology image analysis tasks. In addition, the decoder architecture proposed in this study is flexible and can be incorporated into other deep CNNs for histopathology image analysis. However, we are yet to conduct experiments using natural images; therefore the superiority of our approach in this context cannot be guaranteed. We consider this an open problem and plan to conduct further research to provide a theoretical analysis with complete proof. 

## Figures and Tables

**Figure 1 bioengineering-10-00957-f001:**
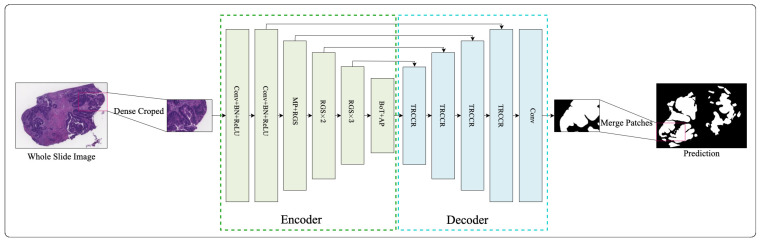
An overview of RGSB-UNet. The TRCCR denotes transposed convolution, ReLU, concatenate, convolution, and ReLU.

**Figure 2 bioengineering-10-00957-f002:**
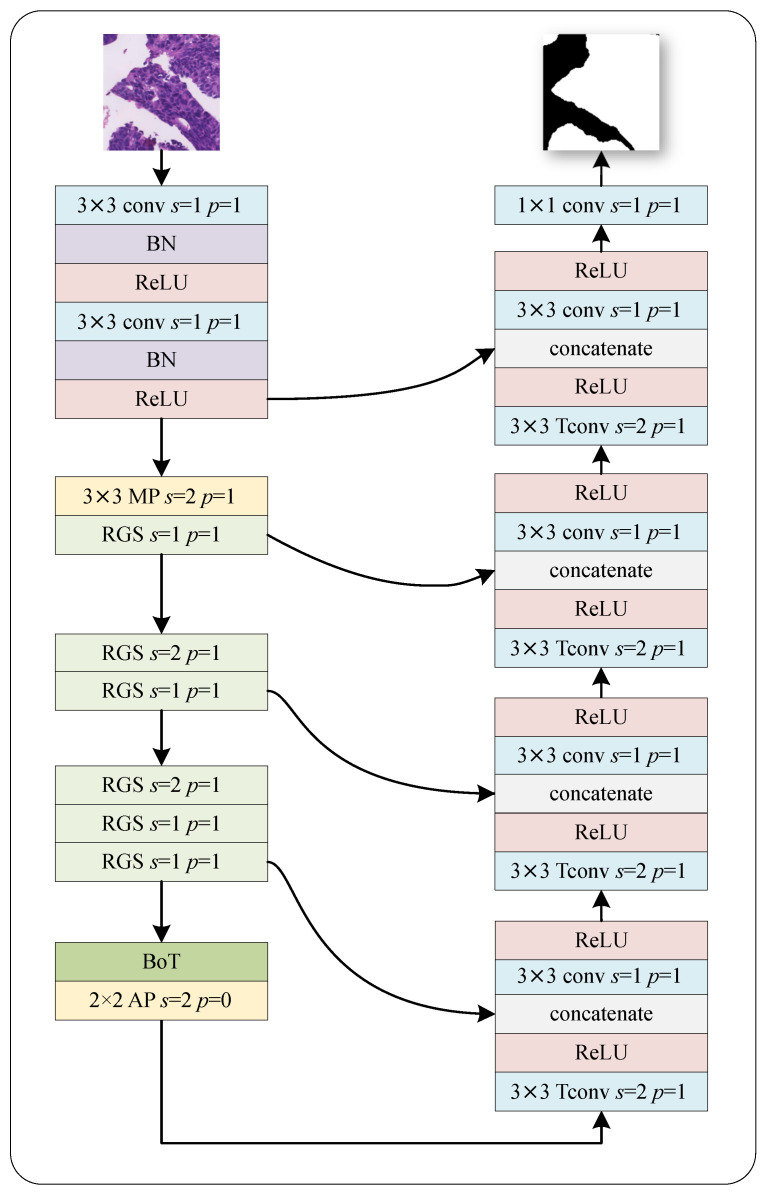
Schematic diagram of RGSB-UNet. RGS denotes the proposed residual ghost block with switchable normalization, and BoT denotes the bottleneck transformer. MP and AP denote the max and average pooling, respectively. Tconv denotes the transposed convolution used for upsampling.

**Figure 3 bioengineering-10-00957-f003:**
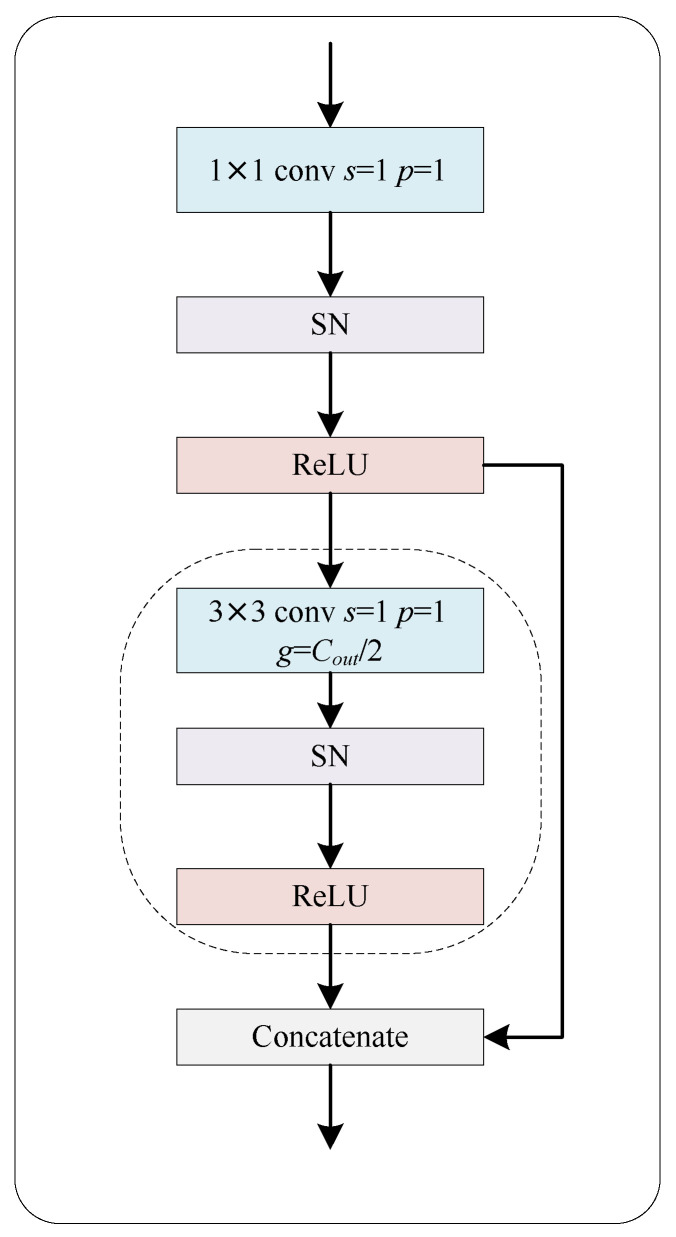
Schematic diagram of Ghost block with switchable normalization. The dash box denotes the cheap operation that uses a 3 × 3 group convolution in the ghost block.

**Figure 4 bioengineering-10-00957-f004:**
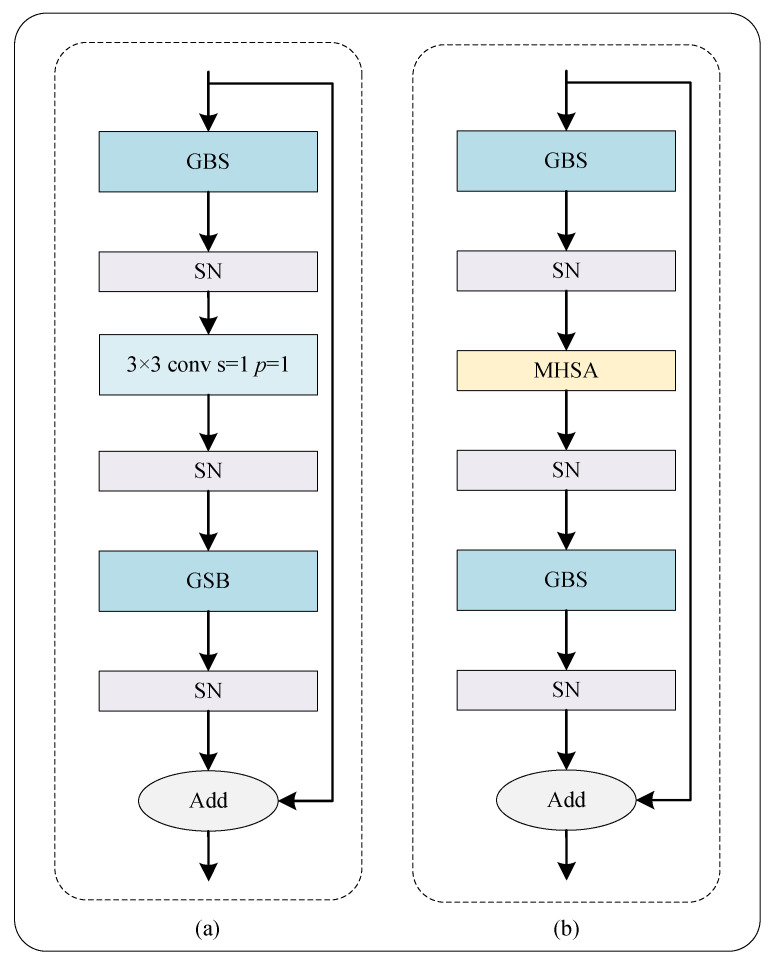
Schematic diagram of the proposed bottleneck. (**a**) RGS Bottleneck. (**b**) Bottleneck transformer. GBS and SN denote the ghost block with switchable normalization and switchable normalization, respectively. MHSA denotes multi-head self-attention.

**Figure 5 bioengineering-10-00957-f005:**
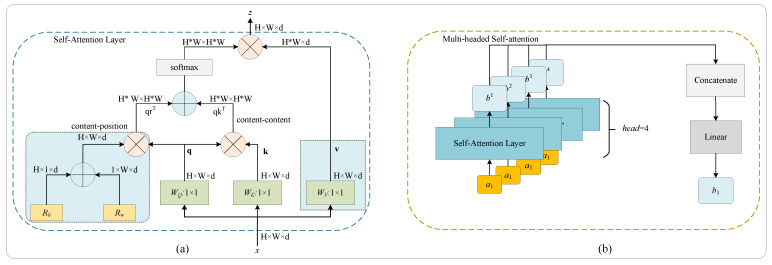
Schematic diagram of (**a**) self-attention [26] and (**b**) multi-head self-attention.

**Figure 6 bioengineering-10-00957-f006:**
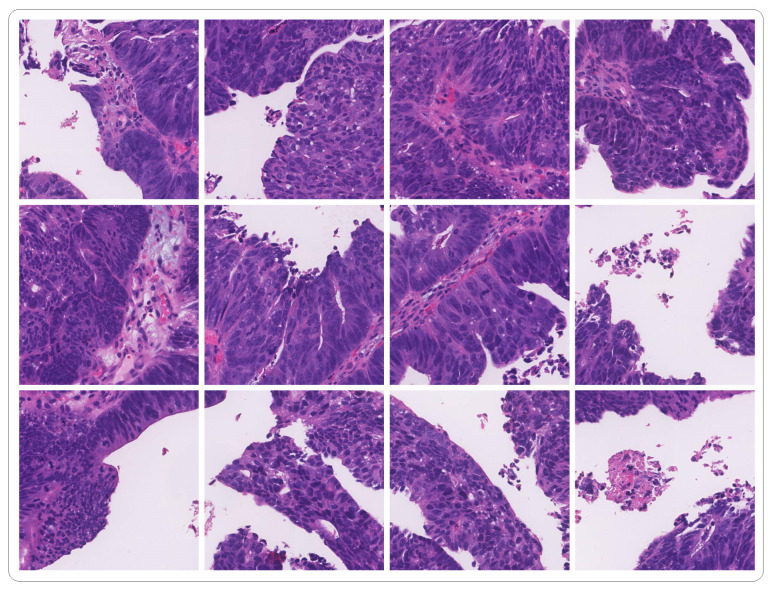
Samples cropped from WSI.

**Figure 7 bioengineering-10-00957-f007:**
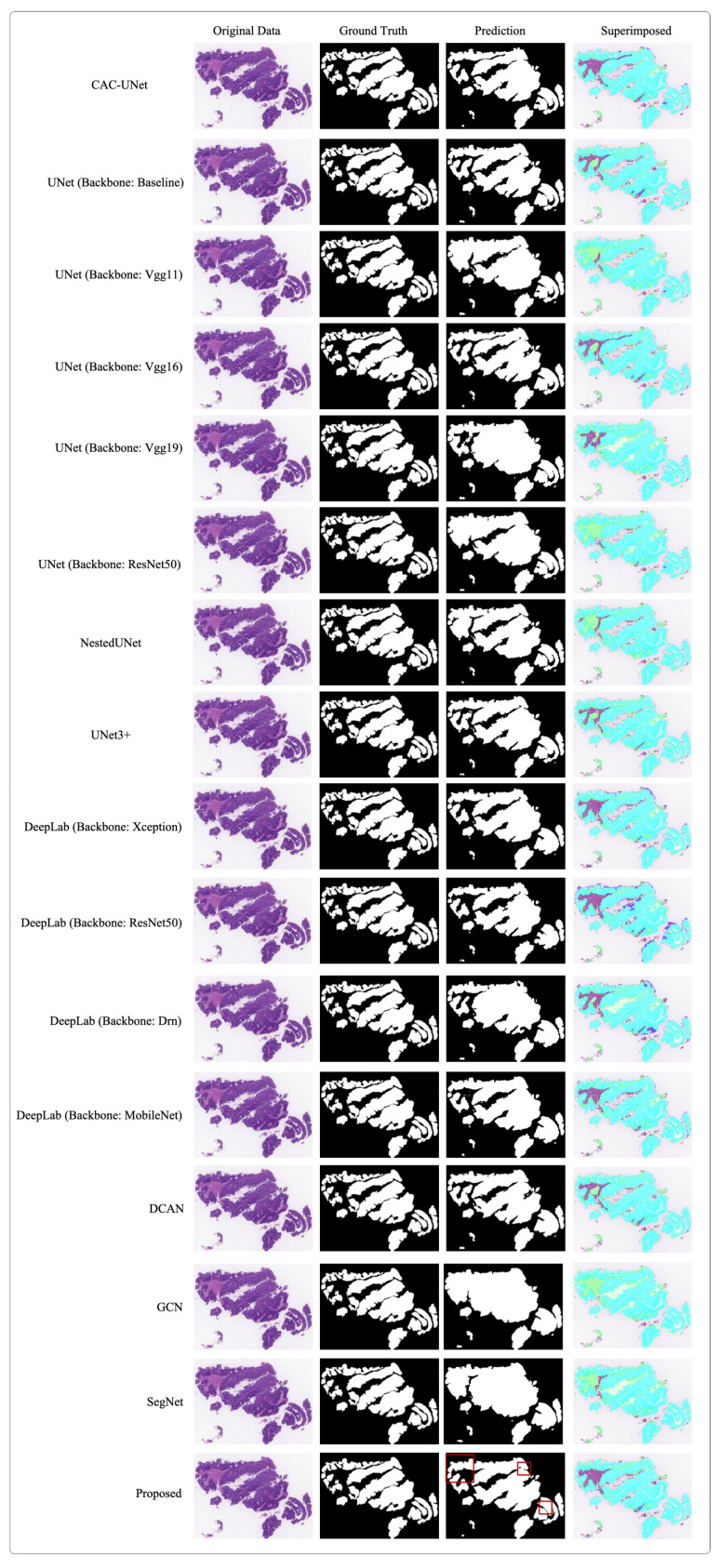
Segmentation results of different networks on the DigestPath2019 dataset. In the superimposed images, the areas marked in green represent the ground truth.

**Figure 8 bioengineering-10-00957-f008:**
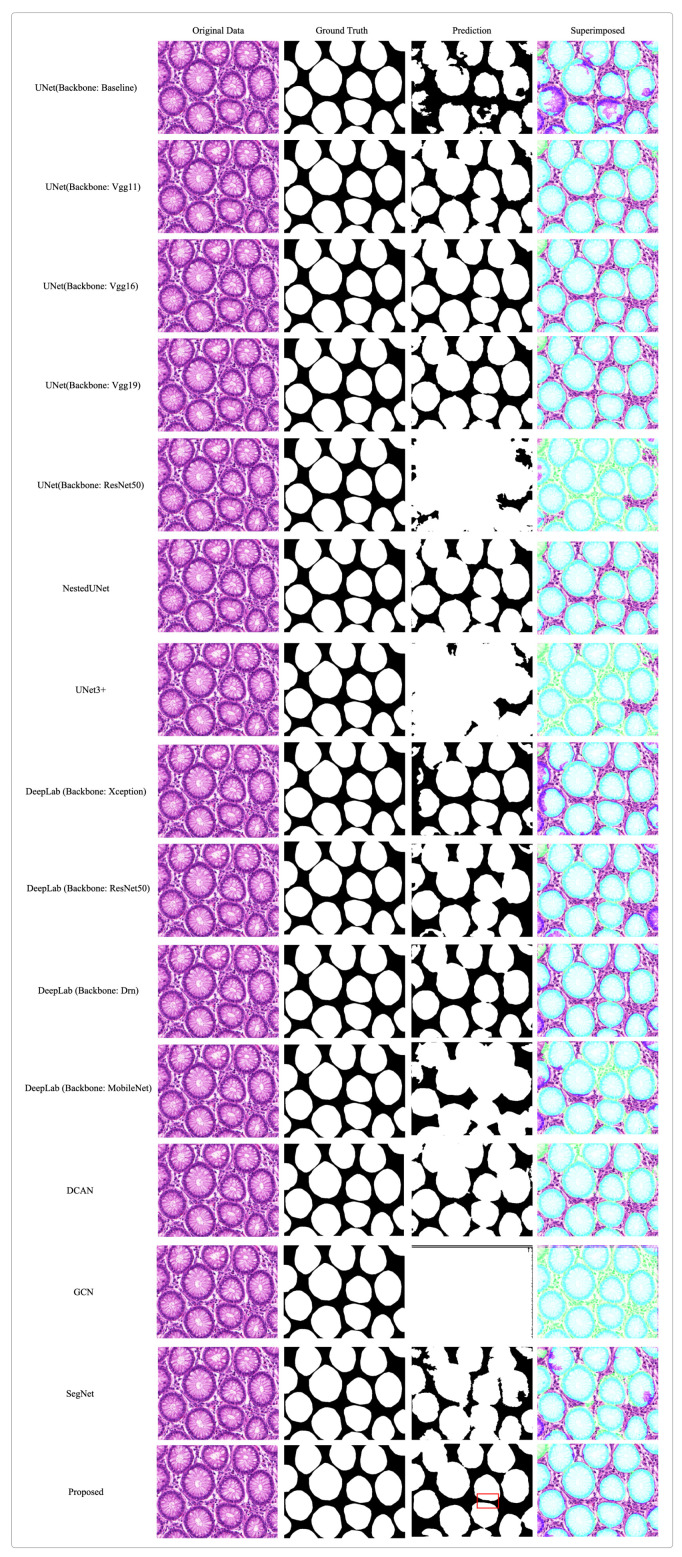
Segmentation results of different networks on the GlaS dataset. In the superimposed images, the areas marked in green represent the ground truth.

**Table 1 bioengineering-10-00957-t001:** Hyperparameters of our framework.

Hyperparameters	Value
Crop Method	Dense Crop
Crop Stride	512
Crop Patch Size	512×512×3
Batch Size	2
MHSA Head	4
Optimizer	SGD
Learning Rate	$1.0\times e^{-2}$
Weight Deacy	$1.0\times e^{-4}$
Momentum	0.9
Epoch Number	500

**Table 2 bioengineering-10-00957-t002:** Performance gains by integrating different blocks into UNet on the DigestPath2019 dataset. RSB and RGB denote the residual block and residual ghost block with batch normalization, respectively.

UNet	RSB	RGB	RGS	BoT	DSC
✔					0.8150
✔	✔				0.8197
✔	✔			✔	0.8201
✔		✔			0.8203
✔		✔		✔	0.8261
✔			✔		0.8263
✔			✔	✔	0.8336

**Table 3 bioengineering-10-00957-t003:** Effect of batch size and MHSA head on model performance. The best results are marked in bold.

Batch Size	1				2			
MHSA Head	☓	1	2	4	☓	1	2	4
DSC	0.8126	0.8241	0.8220	0.8331	0.8263	0.8294	0.8250	**0.8336**

**Table 4 bioengineering-10-00957-t004:** Comparative results for tumour segmentation on the DigestPath2019 dataset. The best results are marked in bold.

Methods	DSC	AUC	PA	JI	RVD	Precision
CAC-UNet [20]	0.8292	1.0000	0.8935	0.7082	0.3219	0.9072
UNet (Baseline) [37]	0.8150	0.9060	0.8611	0.6914	0.2852	0.6511
UNet (Backbone: Vgg11) [38]	0.8258	0.9187	0.8796	0.7081	0.2964	0.6829
UNet (Backbone: Vgg16) [39]	0.8323	0.9562	0.9351	0.7177	0.2445	0.8000
UNet (Backbone: Vgg19) [21]	0.7417	0.5875	0.3889	0.5990	0.4803	0.2987
UNet (Backbone: ResNet50) [40]	0.8197	0.9312	0.8981	0.7019	0.3652	0.7179
UNet (Backbone: DenseNet121) [41]	0.2183	0.5758	0.5092	0.1441	0.4825	0.3076
NestedUNet [42]	0.7609	0.7625	0.6481	0.6254	0.5561	0.4242
Unet3+ [43]	0.7467	0.6250	0.4450	0.6127	0.3977	0.3181
DeepLab (Backbone: Xception) [44]	0.6999	0.9500	0.9259	0.5517	**0.1925**	0.7778
DeepLab (Backbone: ResNet50) [44]	0.7964	0.6375	0.4629	0.6684	0.3829	0.3255
DeepLab (Backbone: Drn) [44]	0.7917	0.7125	0.5740	0.6605	0.3214	0.3783
DeepLab (Backbone: MobileNet) [44]	0.7943	0.8250	0.7407	0.6658	0.4206	0.5000
DCAN [45]	0.8322	0.9562	0.9351	0.7169	0.2291	0.8000
GCN [46]	0.6372	0.6625	0.5000	0.4903	0.5051	0.3414
SegNet [47]	0.7564	0.7937	0.6944	0.6174	0.5845	0.4590
Proposed	**0.8336**	**0.9813**	**0.9722**	**0.7190**	0.2122	**0.9032**

**Table 5 bioengineering-10-00957-t005:** Comparative results for gland segmentation on the GlaS dataset. The best results are marked in bold.

Methods	DSC	AUC	PA	JI	RVD	Precision
UNet (Baseline) [37]	0.5132	0.4339	0.8125	0.3745	0.4959	0.9285
UNet (Backbone: Vgg11) [38]	0.7486	0.5068	0.9480	0.6195	0.6165	0.9313
UNet (Backbone: Vgg16) [39]	0.7324	0.6328	0.8265	0.6038	0.7378	0.8375
UNet (Backbone: Vgg19) [21]	0.7289	0.5979	0.8975	0.5999	0.7595	0.7928
UNet (Backbone: ResNet50) [40]	0.6511	0.5000	0.9375	0.5065	0.9228	0.9375
UNet (Backbone: DenseNet121) [41]	0.6491	0.5998	0.9263	0.5037	0.9046	0.9261
NestedUNet [42]	0.6003	0.4533	0.8500	0.4651	0.8031	0.9315
Unet3+ [43]	0.6650	0.6725	0.9450	0.5170	0.8459	0.9428
DeepLab (Backbone: Xception) [44]	0.6867	0.4735	0.8875	0.5564	0.4423	0.9342
DeepLab (Backbone: ResNet50) [44]	0.6887	0.4866	0.9125	0.5503	0.5648	0.9358
DeepLab (Backbone: Drn) [44]	0.7367	0.5306	0.9375	0.6039	0.6299	0.9375
DeepLab (Backbone: MobileNet) [44]	0.6839	0.4933	0.9250	0.5410	0.6062	0.9367
DCAN [45]	0.6415	0.6107	0.9177	0.4896	0.9459	0.9370
GCN [46]	0.5696	0.5079	0.6983	0.4220	0.9918	0.7863
SegNet [47]	0.5206	0.5533	0.8625	0.3799	0.3995	0.9445
Proposed	**0.8865**	**0.8920**	**0.9823**	**0.7953**	**0.2128**	**0.9475**

## Data Availability

The data that support the findings of this study are available from the corresponding author upon reasonable request.

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
