# Peer review of "RGSB-UNet: Hybrid Deep Learning Framework for Tumour Segmentation in Digital Pathology Images"

_bioengineering, 2023, doi:10.3390/bioengineering10080957_

Round 1

Reviewer 1 Report

Summary: The authors present a paper titled "RGSB-UNet: Hybrid Deep Learning Framework for Tumour Segmentation in Digital Pathology Images" where they propose a new method for tumor segmentation. The proposed method, RGSB-UNet, is trained and evaluated on the training set of DigestPath2019.

Major Comments:

  1. Performance Gain: The paper claims a performance gain for the proposed method based on dice similarity coefficients and Jaccard index. However, it is mentioned that the improvement is less than 1%. This marginal improvement raises questions about the significance of the proposed method and its practical utility. The authors should provide further analysis and explanation regarding the impact of this improvement.
  2. Comparison with DigestPath2019 Challenge Participants: The paper lacks a comparison of results with the participants of the DigestPath2019 challenge. Comparing the proposed method's performance with other state-of-the-art methods from the challenge would provide a better understanding of its effectiveness. This comparison is essential for evaluating the novelty and competitiveness of the proposed approach.
  3. Generalizability: To establish the significance of the proposed method, it is crucial to validate the results on other digital pathology datasets. The authors should include experiments on datasets such as the Camelyon16-17 challenge or the Gland Segmentation in Colon Histology Images (GLAS) challenge. This would demonstrate the method's generalizability and its performance in different contexts.
  4. Lack of Referencing and Results: The authors claim the importance of various parts of the network architecture without providing any references or results to support those claims. It is important to provide evidence or references to justify the design choices and the significance of each component. This would enhance the credibility and reliability of the proposed method.

The following sentences are disconnected from the context of the para in which they appear.

"The challenges in histopathology image analysis should be effectively addressed with upcoming deep learning methods [8]." 

"A method has successfully leveraged a deep neural network with scribbles for interactive pathology image segmentation [15] and an attention-guided connection module for liver tumour segmentation [16]." 

"Although incorporating contextual information can avoid interference from complex background speckles, it remains insensitive to images with sparse staining owing to network architecture limitations."

Reviewer 2 Report

In this paper, authors proposed a hybrid deep learning framework for automatic tumour segmentation in WSIs. The framework adopts an encoder-decoder architecture that consists of the newly-designed residual ghost block with switchable normalization (RGS) and the bottleneck transformer (BoT) for downsampling to extract refined features, and the transposed convolution and 1×1 convolution with ReLU for upsampling to restore the feature map resolution to that of the original image. The proposed framework combines the advantages of the spatial-local correlation of CNNs and the long-distance feature dependencies of BoT, ensuring its capacity of extracting more refined features and robustness to varying batch sizes. Authors did a good work and interested for the readers. Following review comments are recommended, and the authors are invited to explain and modify.

1 Authors should give full name of “RGSB-UNet” in abstract.

2 “Our novel hybrid backbone encoder, which includes RGS and BoT blocks, can extract more refined features”, what is novelty?

3 Introduction section needs to be improve with latest references, no any ref was found of year 2023. An introduction is an important road map for the rest of the paper that should be consist of an opening hook to catch the researcher's attention, relevant background study, and a concrete statement that presents main argument but your introduction lacks these fundamentals, especially relevant background studies. This related work is just listed out without comparing the relationship between this paper's model and them; only the method flow is introduced at the end; and the principle of the method is not explained. To make soundness of your study must include these latest related works. Authors also need to justify the importance of their article and cite all of them to make a critical discussion that makes a difference from others' work.

I (2023). Interpretable Multi-Modal Image Registration Network Based on Disentangled Convolutional Sparse Coding. IEEE Transactions on Image Processing, 32, 1078-1091. doi: 10.1109/TIP.2023.3240024

II (2023). Interpretable Multi-Modal Image Registration Network Based on Disentangled Convolutional Sparse Coding. IEEE Transactions on Image Processing, 32, 1078-1091. doi: 10.1109/TIP.2023.3240024

III (2023). iERM: An Interpretable Deep Learning System to Classify Epiretinal Membrane for Different Optical Coherence Tomography Devices: A Multi-Center Analysis. Journal of Clinical Medicine, 12(2). doi: 10.3390/jcm12020400

IV (2022). DCGNN: a single-stage 3D object detection network based on density clustering and graph neural network. Complex & Intelligent Systems. doi: 10.1007/s40747-022-00926-z

4 Methodology section should have a detailed flowchart of the whole work. This will help the reader to get a better understanding of what is going on in the proposed ‎system.‎

5 “We selected 108 images, from 660 negative and positive images,” what was selection criteria and how to balance it?

6 There are some of typos. English needs to revise again, see: “Making our model robust robust to varying batch sizes”,

7 When writing phrases like “We uses the DSC, Jaccard Index (JI), and relative volume difference (RVD) to measure the segmentation performance of our proposed model”, it must cite related work in order to sustain the statement https://doi.org/10.1155/2023/2345835.

8 “Table 1. Hyperparameters of our framework:, how to optimize hyperparameters during model training?

9 Authors should mention the implementation challenges.

Moderate editing of English language required.

Round 2

Reviewer 1 Report

I want to express my sincere appreciation for addressing the major comments I raised during the first revision of your paper. Your efforts have significantly improved the manuscript. However, to ensure future readers understand your work better, please add further information to clarify the following points regarding your experimental setup.

In Table 4, you presented the results for the CAC-UNet method, but these reported results differ from those found in the cited paper. To gain a clearer understanding, could you please elaborate on how you obtained these results? Specifically, explain whether you retrained the CAC-UNet model on your dataset. If yes, then whether you used the original CAC-UNET code or implemented their method in your experiments. Additionally, kindly clarify whether you used the same test set as the DigestPath2019 dataset, where the results of various methods are publicly available. 

In Table 5, I noticed that your method's performance on the GLAS Challenge Dataset is comparable to that of the top-performing method. To provide a comprehensive understanding of your approach, please provide details on the following points:

  1. The number of samples you used for the train, valid, and test sets in your experimental setup.
  2. Clarification on how the optimal model was selected. Did you use the validation set or the test set for this purpose?
  3. Please confirm whether your test set is the same as the test set used for the GLAS dataset, considering that the GLAS dataset offers two different test sets.

Including these additional details will enhance the reproducibility of your work and contribute to a more comprehensive evaluation of your contributions by the research community.

Reviewer 2 Report

Authors have answered all my questions satisfactorily.

Author Response

Thanks for the reviewer’s valuable suggestions and effort to improve the manuscript.

Round 3

Reviewer 1 Report

Thanks for adding the required information for completeness.